# ANGLE-OPTIMIZED TEXT EMBEDDINGS

## ABSTRACT

High-quality text embedding is pivotal in improving semantic textual similarity (STS) tasks, which are crucial components in Large Language Model (LLM) applications. However, a common challenge existing text embedding models face is the problem of vanishing gradients, primarily due to their reliance on the cosine function in the optimization objective, which has saturation zones. To address this issue, this paper proposes a novel angle-optimized text embedding model called AnglE. The core idea of AnglE is to introduce angle optimization in a complex space. This novel approach effectively mitigates the adverse effects of the saturation zone in the cosine function, which can impede gradient and hinder optimization processes. To set up a comprehensive STS evaluation, we experimented on existing short-text STS datasets and a newly collected long-text STS dataset from GitHub Issues. Furthermore, we examine domain-specific STS scenarios with limited labeled data and explore how AnglE works with LLM-annotated data. Extensive experiments were conducted on various tasks including short-text STS, long-text STS, and domain-specific STS tasks. The results show that AnglE outperforms the state-of-the-art (SOTA) STS models that ignore the cosine saturation zone. These findings demonstrate the ability of AnglE to generate high-quality text embeddings and the usefulness of angle optimization in STS.

## 1 INTRODUCTION

The development of text embeddings (Kiros et al., 2015; Hill et al., 2016; Conneau et al., 2017; Cer et al., 2018; Reimers & Gurevych, 2019; Gao et al., 2021) is an essential research challenge in the NLP community. Text embeddings effectively feature key semantic and syntactic information in language, which broadly affects the performance of downstream tasks, such as text classification (Li et al., 2021), sentiment analysis (Suresh & Ong, 2021; Zhang et al., 2022), semantic matching (Grill et al., 2020; Lu et al., 2020), clustering (Reimers & Gurevych, 2019; Xu et al., 2023), and question-answering (QA) system (Yue et al., 2021). In particular, text embedding models play a crucial role in LLMs such as ChatGPT (OpenAI, 2022; 2023), LLaMA (Touvron et al., 2023a;b), and ChatGLM (Du et al., 2022)-based applications. These LLM-based applications heavily rely on high-quality text embeddings for tasks such as vector search, where related documents are retrieved for LLM QA (Asai et al., 2023).

Recent studies (Gao et al., 2021; Jiang et al., 2022; Chuang et al., 2022; Chanchani & Huang, 2023; Zhuo et al., 2023) have utilized pre-trained language models such as BERT (Devlin et al., 2019) and RoBERTa (Liu et al., 2019) in combination with contrastive learning to enhance the quality of text embeddings. These approaches involve pulling semantically similar samples together and pushing apart those not (Gao et al., 2021). In these contrastive models, positive samples that are semantically similar can be generated by data augmentation, while negative samples that are dissimilar are selected from different texts within the same mini-batch (in-batch negatives). However, supervised negatives are underutilized, and the correctness of in-batch negatives is difficult to guarantee without annotation, which can lead to performance degradation. Although some models such as (Gao et al., 2021) optimize hard negative samples, they rely on strict triple formats $(x_i, x_i^+, x_i^-)$. While most existing supervised STS datasets only provide pairs $(x_i, x_i^+)$ or $(x_j, x_j^-)$, where $x_i^+$ refers to the positive sample of $x_i$ while $x_j^-$ the negative sample of $x_j$. Thus, most contrastive models are used in unsupervised settings yet might not benefit from human supervision.

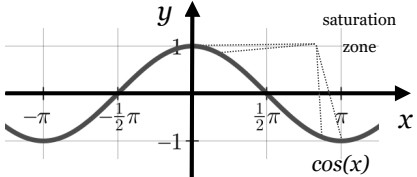

Figure 1: The saturation zones of the cosine function. The gradient at saturation zones is close to zero. During backpropagation, if the gradient is very small, it could kill the gradient and make the network difficult to learn.

For supervised STS (Reimers & Gurevych, 2019; Su, 2022), most efforts to date employed the cosine function in their training objective to measure the pairwise semantic similarity. However, the cosine function has saturation zones, as shown in Figure 1. The functions with saturation zones, such as sigmoid and tanh, can lead to the problem of gradient vanishing (Roodschild et al., 2020), regardless of the depth of the network. This means that if two points $x_1$ and $x_2$ fall within the saturation zone, the outputs $f(x_1)$ and $f(x_2)$ will be very close to each other. As a result, the gradient $\Delta = f(x_1) - f(x_2) \approx 0$ will vanish during backpropagation. It can impede the optimization and hinder the ability to learn subtle differences between texts in backpropagation. Many STS datasets, such as MNLI (Williams et al., 2018) and SNLI (Bowman et al., 2015), provide three supervised labels: entailment, neutral, and contradiction. In most cases, the boundary between neutral and entailment is vague, as some neutral pairs may resemble entailment pairs. For instance, consider the following cases from SNLI: (1) the text pair of *Two blond women hug each other.* and *Some women hug each other on vacation.* are labeled as neutral, and (2) the text pair of *Two blonde women hugging.* and *There are women showing affection.* are labeled as entailment. The texts in the neutral pair appear very similar to each other and may fall within the saturation zone of cosine, resulting in gradient vanishing. This can confuse the model and lead to misidentification of the neutral pair as entailment. Additionally, many other STS datasets such as MRPC [1] and QQP [2] provide binary labels representing dissimilar (0) and similar (1), which naturally fall within the saturation zone of the cosine function. To overcome these challenges, this paper proposes a novel angle-optimized text embedding. It optimizes not only the cosine similarity between texts but also the angle to mitigate the negative impact of the saturation zones of the cosine function on the learning process. Specifically, it first divides the text embedding into real and imaginary parts in a complex space. Then, it follows the division rule in complex space to compute the angle difference between two text embeddings. After normalization, the angle difference becomes an objective to be optimized. There are also some works (Meng et al., 2019) optimizing loss in complicated spherical manifolds. However, these works tend to be complicated and have higher complexity, making them inefficient. Efficiency is essential for text embedding learning, especially in the era of LLMs. Because LLMs have large-scale parameters and are hard to be finetuned. Compared to them, optimizing the angle difference in complex space is more efficient and practical. It is also more intuitive because if the angle difference between two text embeddings is smaller, it means that the two text embeddings are closer to each other in the complex space, i.e., their similarity is larger.

In the STS experimental setup, we observed that the majority of existing STS benchmarks focus on evaluating models on short texts. Unfortunately, there is a lack of datasets specifically designed to evaluate the STS performance of models on long texts. Long texts are prevalent in real-world applications such as financial documents, legal documents, and health reports (Li et al., 2023). To tackle this challenge, this paper presents a new high-quality long-text STS dataset. This dataset allows for a more thorough evaluation of model performance on long texts. Specifically, the dataset is collected from GitHub Issues with roughly 22K samples, we use the duplicate issues as the positive samples and the non-duplicate issues as the negative samples.

We first experimented with both short and long-text datasets and showed that AnglE outperforms the SOTA STS models in both transfer and non-transfer STS tasks. For example, AnglE shows an average Spearman correlation of $73.55\%$ in non-transfer STS tasks, compared to $68.03\%$ for SBERT. Then, an ablation study shows that all components contribute positively to the superior performance of AnglE. Next, we discuss the domain-specific scenarios with limited annotated data that are challenging for AnglE-like supervised STS, where it is observed that AnglE can work well with LLM-supervised data. Finally, we find that AnglE can benefit downstream retrieval applications and can learn representations closer to actual representations.

---

[1] https://www.microsoft.com/en-us/download/details.aspx?id=52398

[2] https://www.quora.com/q/quoradata/

In summary, the contributions of this paper are as follows: 1) We investigate the negative effects of saturation zone in the cosine function widely applied in STS tasks and propose a novel angle-optimized text embedding model to mitigate this issue. 2) We extend the existing STS benchmark with a newly collected long-text dataset from Github Issues to allow a more comprehensive empirical study in STS. 3) We present extensive experiments on STS which demonstrate that AnglE outperforms SOTAs and can substantially improve the text embedding quality in various scenarios.

## 2 RELATED WORK

This section is organized as follows: we first introduce the unsupervised approaches, then the supervised approaches, and finally give a summary.

**Unsupervised Approaches** Early studies (Hill et al., 2016; Pagliardini et al., 2018) have demonstrated the efficacy of augmenting word2vec (Mikolov et al., 2013) with n-gram embeddings, yielding strong results in text embeddings. Recently, BERT-flow (Li et al., 2020) has introduced a flow-based approach that maps BERT embeddings to a standard Gaussian latent space. On the other hand, BERT-whitening (Su et al., 2021) applies the whitening operation to BERT embeddings to enhance text embeddings. Furthermore, very recent research (Carlsson et al., 2020; Zhang et al., 2020; Giorgi et al., 2021; Gao et al., 2021; Yan et al., 2021; Chuang et al., 2022; Jiang et al., 2022; Zhuo et al., 2023) has focused on leveraging contrastive objectives to improve the quality of text embeddings.

**Supervised Approaches** Supervised text embeddings usually perform better than their unsupervised counterparts (Gao et al., 2021). Various studies have effectively utilized supervised datasets to enhance the learning of text embeddings. In particular, Conneau et al. (2017) introduced a method that leverages supervised Natural Language Inference (NLI) tasks for this purpose. Building on a transformer backbone, USE (Cer et al., 2018) incorporates the SNLI dataset to augment unsupervised training, resulting in improved performance. Furthermore, SBERT (Reimers & Gurevych, 2019) enhances text embedding by combining BERT with a siamese architecture.

However, most existing models optimize the cosine similarity, as shown in the appendix section A.1, but neglect the negative effect of the saturation zones in cosine. To address this issue, this paper proposes a novel angle-optimized text embedding model to improve the quality of text embedding.

## 3 METHODOLOGY

This section will introduce the components of the proposed angle-optimized text embedding model, including the input layer, cosine objective, in-batch negative objective, and angle objective.

### 3.1 INPUT LAYER

For the input sentences, we first apply padding to ensure a consistent length $l$. Next, we map each word to a continuous $d$-dimensional space to produce word embeddings $\mathbf{e}_i \in \mathbb{R}^d$. These word embeddings are then concatenated to form the model input: $\mathbf{E} = [\mathbf{e}_1, \mathbf{e}_2, \dots, \mathbf{e}_l] \in \mathbb{R}^{l \times d}$. Subsequently, the model input is passed through an encoder such as BERT (Devlin et al., 2019), RoBERTa (Liu et al., 2019), and LLaMA (Touvron et al., 2023a;b) to obtain the contextual representation $\mathbf{X}$.

### 3.2 COSINE OBJECTIVE

Following the prior study (Su, 2022), we employ the cosine objective function for end-to-end optimization of cosine similarity between representations, as follows:

$$\mathcal{L}_{cos} = \log \left[ 1 + \sum_{s(\mathbf{X}_i, \mathbf{X}_j) > s(\mathbf{X}_m, \mathbf{X}_n)} e^{\frac{\cos(\mathbf{X}_m, \mathbf{X}_n) - \cos(\mathbf{X}_i, \mathbf{X}_j)}{\tau}} \right], \quad (1)$$

where $\tau$ is a temperature hyperparameter, $\cos(\cdot)$ is the cosine similarity function, and $s(u, v)$ is the similarity between $u$ and $v$. By optimizing the $\mathcal{L}_{cos}$, we expect the cosine similarity of the high similarity pair to be greater than that of the low similarity pair.

### 3.3 IN-BATCH NEGATIVE OBJECTIVE

To further improve performance, we integrate the in-batch negative objective function. Because in-batch negative samples can serve as a data augmentation technique, which can strengthen the ability to generalize. Unlike existing contrastive learning methods (Gao et al., 2021; Yan et al., 2021; Jiang et al., 2022) that generate positive samples through data augmentation, we use supervised positive samples. Recognizing that there might be identical sentences within a batch that are not explicitly labeled as positive samples, causing them to become in-batch negatives, we identify these duplicate sentences and assign them as positive samples, thereby reducing potential noise. The formulation for the in-batch negative objective function (ibn) is as follows:

$$\mathcal{L}_{ibn} = -\sum_b \sum_i^m \log \left[ \frac{e^{\cos(\mathbf{X}_{b_i}, \mathbf{X}_{b_i}^+)/\tau}}{\sum_j^N e^{\cos(\mathbf{X}_{b_i}, \mathbf{X}_{b_j}^+)/\tau}} \right], \tag{2}$$

where $\tau$ is a temperature hyperparameter, $b$ stands for the $b$-th batch, $\mathbf{X}_{b_i}^+$ and $\mathbf{X}_{b_j}^+$ are the respective positive samples of $\mathbf{X}_{b_i}$ and $\mathbf{X}_{b_j}$, $m$ represents the number of positive pairs in $b$-th batch, $N$ is the batch size, and $\cos(\cdot)$ is the cosine similarity function.

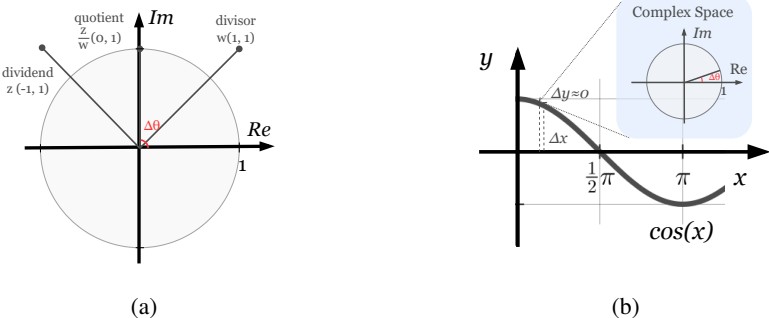

(a)            (b)

Figure 2: (a) Division in complex space. $\Delta\theta$ is the angle difference between dividend $z$ and divisor $w$ in complex space. (b) Angle optimization in cosine saturation zones. Even though $\Delta y \approx 0$ could kill the gradient, the corresponding angle difference in complex space is still distinct for optimization.

### 3.4 ANGLE OBJECTIVE

We found that both the cosine and in-batch negative objectives employ the cosine function to measure similarity. However, it is important to note that the cosine function includes saturation zones, which can hinder the optimization process. We optimize the angle difference in complex space to mitigate these adverse effects. Figure 2a draws the division in complex space and Figure 2b depicts how angle optimization works in cosine saturation zones. To optimize the angle difference, we initially decompose the representation $\mathbf{X}$ into its real part $\mathbf{X}^{re}$ and imaginary part $\mathbf{X}^{im}$. For the pair $(\mathbf{X}_i, \mathbf{X}_j)$, their representations in the complex space are defined as follows:

$$\begin{aligned} \mathbf{z} &= \mathbf{a} + \mathbf{b}i \in \mathbb{C} \\ \mathbf{w} &= \mathbf{c} + \mathbf{d}i \in \mathbb{C}, \end{aligned} \tag{3}$$

where $\mathbf{a} = \mathbf{X}_i^{re} \in \mathbb{R}$, $\mathbf{b} = \mathbf{X}_i^{im} \in \mathbb{R}$, $\mathbf{c} = \mathbf{X}_j^{re} \in \mathbb{R}$, and $\mathbf{d} = \mathbf{X}_j^{im} \in \mathbb{R}$. To compute the angle difference between $\mathbf{z}$ and $\mathbf{w}$, we calculate division in complex space in polar coordinates, as follows:

$$\begin{aligned} \frac{\mathbf{z}}{\mathbf{w}} &= \gamma \Delta\theta_{zw} \\ \gamma &= \frac{r_{\mathbf{z}}}{r_{\mathbf{w}}} = \frac{\sqrt{\mathbf{a}^2 + \mathbf{b}^2}}{\sqrt{\mathbf{c}^2 + \mathbf{d}^2}} \\ \Delta\theta_{zw} &= \theta_{\mathbf{z}} - \theta_{\mathbf{w}}, \end{aligned} \tag{4}$$

where $r_{\mathbf{z}}$ and $r_{\mathbf{w}}$ represent the magnitudes of $\mathbf{z}$ and $\mathbf{w}$, while $\theta_{\mathbf{z}}$ and $\theta_{\mathbf{w}}$ denote the respective angles of $\mathbf{z}$ and $\mathbf{w}$. For a detailed derivation of this complex space division, please refer to Appendix A.2.

Next, we compute the value of $\frac{\mathbf{z}}{\mathbf{w}}$ by the division rule in complex space (see A.3), as follows:

$$\frac{\mathbf{z}}{\mathbf{w}} = \frac{\mathbf{a} + \mathbf{b}i}{\mathbf{c} + \mathbf{d}i} = \frac{(\mathbf{ac} + \mathbf{bd}) + (\mathbf{bc} - \mathbf{ad})i}{\mathbf{c}^2 + \mathbf{d}^2}. \tag{5}$$

By employing Eq. 4 and Eq. 5, we can calculate the angle difference between $\mathbf{z}$ and $\mathbf{w}$ by multiplying both sides by $\frac{1}{\gamma}$, which can be seen as a normalization operation. In this paper, we determine the absolute normalized angle difference using the following expression:

$$\begin{aligned}
\Delta\theta_{zw} &= \mathrm{abs}(\frac{\mathbf{z}}{\mathbf{w}} \times \frac{1}{\gamma}) \\
&= \mathrm{abs}\left[\frac{(\mathbf{ac} + \mathbf{bd}) + (\mathbf{bc} - \mathbf{ad})i}{\mathbf{c}^2 + \mathbf{d}^2} \times \frac{\sqrt{\mathbf{c}^2 + \mathbf{d}^2}}{\sqrt{\mathbf{a}^2 + \mathbf{b}^2}}\right] \\
&= \mathrm{abs}\left[\frac{(\mathbf{ac} + \mathbf{bd}) + (\mathbf{bc} - \mathbf{ad})i}{\sqrt{(\mathbf{c}^2 + \mathbf{d}^2)(\mathbf{a}^2 + \mathbf{b}^2)}}\right].
\end{aligned} \tag{6}$$

The detailed implementation of the angle difference is presented in Section A.6. Then, the angle difference can be optimized by the following objective function:

$$\mathcal{L}_{angle} = \log\left[1 + \sum_{s(\mathbf{X}_i, \mathbf{X}_j) > s(\mathbf{X}_m, \mathbf{X}_n)} e^{\frac{\Delta\theta_{ij} - \Delta\theta_{mn}}{\tau}}\right], \tag{7}$$

where $\tau$ is a temperature hyperparameter and $s(u, v)$ is the similarity between $u$ and $v$. By optimizing the $\mathcal{L}_{angle}$, our objective is to minimize the normalized angle difference for pairs with high similarity compared to those with low similarity.

Finally, we combine the aforementioned three objective functions in the following manner to form the final objective function:

$$\mathcal{L} = w_1 * \mathcal{L}_{cos} + w_2 * \mathcal{L}_{ibn} + w_3 * \mathcal{L}_{angle}, \tag{8}$$

where $w_1$, $w_2$, and $w_3$ are constants.

## 4 EXPERIMENT

### 4.1 DATASETS AND EVALUATION METRICS

**Existing STS Benchmarks**  We mainly evaluate our model on several widely-adopted STS datasets, namely: MRPC, QQP, QNLI [3], STS 2012-2016 (Agirre et al., 2012; 2013; 2014; 2015; 2016), SICK-R (Marelli et al., 2014), and STS-B (Cer et al., 2017). These datasets mainly consist of short text, but real-world scenarios often involve long text documents. Thus, we introduce a newly long-text dataset called *GitHub Issues Similarity Dataset* to evaluate the STS task comprehensively.

**GitHub Issues Similarity Dataset**  We observed the presence of many duplicate issues on GitHub. Typically, the maintainers of open source organizations tend to mark these duplicate issues as closed with a comment like "closing as a duplicate of #id". Consequently, these duplicate issues inherently serve as a source of the STS task. It is also worth noting that most issues contain long texts because of the inclusion of extensive code within the issues. To compile the dataset, we extracted duplicated issues from 55 popular open-source projects (see A.4) on GitHub using GitHub API [4]. The duplicated issues were used as positive samples, while the remaining issues were considered negative samples. Table 1 presents statistics of the GitHub Issues Similarity Dataset, while Figure 3 shows a violin plot illustrating the token-level text length distribution. The visualization reveals a substantial number of long texts. Specifically, the proportion of long texts (token length $> 512$) for the train, validation, and test sets is at $61.03\%$, $60.85\%$, and $60.50\%$, respectively.

**Evaluation Metrics**  To ensure a fair comparison, we follow previous studies and use Spearman's correlation for evaluation. We use SentEval (Conneau & Kiela, 2018) to compute Spearman's correlation and report the results in the "all" setting, which is consistent with the baselines.

---

[3]https://gluebenchmark.com/
[4]https://docs.github.com/en/rest

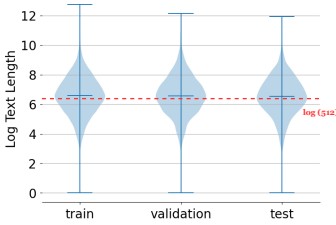

Figure 3: Log token length distribution of the GitHub Issue Similarity Dataset.

Table 1: Statistics of the proposed GitHub Issues Similarity Dataset. #Positive denotes the count of positive pairs, and #Negative represents the number of negative pairs.

| Split | Train | Validation | Test |
|---|---|---|---|
| #Positive | 9457 | 774 | 807 |
| #Negative | 9108 | 773 | 741 |
| Total | 18565 | 1547 | 1548 |

## 4.2 IMPLEMENTATION DETAILS

In this paper, we use the pre-trained uncased BERT base model (110M parameters) as the backbone model. For a fair comparison, all BERT-based baselines also adopt this setting. We set the value of $\tau$ for the cosine objective and the in-batch negative objective to $0.05$, based on prior research. Additionally, we determined the value of $\tau$ for the angle objective to be $1.0$ through grid search. For detailed hyperparameter settings, see A.5.

## 4.3 MAIN RESULTS

In this section, we will first introduce the baselines, then the results of the transfer STS tasks, then the results of the non-transfer STS tasks, and finally a summary.

**Baselines** We compare our proposed model with widely used baselines, encompassing both unsupervised and supervised models. The unsupervised models are average GloVe (Pennington et al., 2014), BERT-flow (Li et al., 2020), BERT-whitening (Su et al., 2021), LLaMA2 (Touvron et al., 2023b), and contrastive learning models including IS-BERT (Zhang et al., 2020), CT-BERT (Carlsson et al., 2020), SimCSE (Gao et al., 2021), ConSERT (Yan et al., 2021), and DiffCSE (Chuang et al., 2022). On the other hand, the chosen supervised models are InferSent (Conneau et al., 2017), USE (Cer et al., 2018), SBERT (Reimers & Gurevych, 2019), CoSENT (Su, 2022), as well as supervised versions of SimCSE and ConSERT.

**Transfer STS Tasks** For a fair comparison, we train AnglE with the NLI datasets MNLI (Williams et al., 2018) and SNLI (Bowman et al., 2015) and then transfer it to evaluate seven STS benchmark datasets. The evaluation results are presented in Table 2. From the table, we can see that AnglE achieves state-of-the-art performance on all STS datasets. It is evident that AnglE-BERT and AnglE-LLaMA consistently outperform the baselines with a gain of $0.80\%$ and $0.72\%$ in average score, respectively, over the previous SOTA SimCSE-BERT and SimCSE-LLaMA. Note that supervised SBERT and CoSENT show lower results than other unsupervised contrastive learning models like SimCSE and DiffCSE. This difference might arise from the difference in data distributions between the training and test data in the transfer STS tasks. They struggle to effectively generalize to STS tasks when trained solely with NLI datasets. In contrast, contrastive learning models exhibit better generalization capabilities due to their alignment and uniformity features. Because AnglE optimizes both the supervised cosine objective and the in-batch negative objective. This can allow AnglE to generalize well in transfer STS tasks. Additionally, the angle optimization in AnglE mitigates the negative impact of the saturation zone in the cosine function to produce better performance than other baseline models.

**Non-transfer STS Tasks** To provide a comprehensive analysis, we also evaluate the performance of the baselines in the non-transfer setting. We train the baselines on the train set and evaluate them on the test or validation set. Two typical models, SimCSE and SBERT, representing contrastive and supervised learning, are compared with our model. The results of the non-transfer STS tasks are listed in Table 3, where we evaluate the baselines on four short-text datasets (MRPC, STS-B, QQP, and QNLI) and one long-text dataset (GitHub Issues Similarity Dataset). SimCSE notably performs poorly compared to SBERT and AnglE in the non-transfer setting. This is due to the limitation of

| Model | STS12 | STS13 | STS14 | STS15 | STS16 | STS-B | SICR-R | Avg. |
|---|---|---|---|---|---|---|---|---|
| *Unsupervised Models* | | | | | | | | |
| GloVe (avg.) † | 55.14 | 70.66 | 59.73 | 68.25 | 63.66 | 58.02 | 53.76 | 61.32 |
| BERT-flow ‡ | 58.40 | 67.10 | 60.85 | 75.16 | 71.22 | 68.66 | 64.47 | 66.55 |
| BERT-whitening ‡ | 57.83 | 66.90 | 60.90 | 75.08 | 71.31 | 68.24 | 63.73 | 66.28 |
| IS-BERT ‡ | 56.77 | 69.24 | 61.21 | 75.23 | 70.16 | 69.21 | 64.25 | 66.58 |
| CT-BERT ‡ | 61.63 | 76.80 | 68.47 | 77.50 | 76.48 | 74.31 | 69.19 | 72.05 |
| ConSERT-BERT | 64.64 | 78.49 | 69.07 | 79.72 | 75.95 | 73.97 | 67.31 | 72.74 |
| DiffCSE-BERT | 72.28 | 84.43 | 76.47 | 83.90 | 80.54 | 80.59 | 71.23 | 78.49 |
| SimCSE-BERT | 68.40 | 82.41 | 74.38 | 80.91 | 78.56 | 76.85 | 72.23 | 76.25 |
| LLaMA2-7B ⋆ | 50.66 | 73.32 | 62.76 | 67.00 | 70.98 | 63.28 | 67.40 | 65.06 |
| *Supervised Models* | | | | | | | | |
| InferSent-GloVe † | 52.86 | 66.75 | 62.15 | 72.77 | 66.87 | 68.03 | 65.65 | 65.01 |
| USE † | 64.49 | 67.80 | 64.61 | 76.83 | 73.18 | 74.92 | 76.69 | 71.22 |
| ConSERT-BERT | 74.07 | 83.93 | 77.05 | 83.66 | 78.76 | 81.36 | 76.77 | 79.37 |
| CoSENT-BERT ⋆ | 71.35 | 77.52 | 75.05 | 79.68 | 76.05 | 78.99 | 71.19 | 75.69 |
| SBERT † | 70.97 | 76.53 | 73.19 | 79.09 | 74.30 | 77.03 | 72.91 | 74.89 |
| SimCSE-BERT | 75.30 | 84.67 | 80.19 | 85.40 | 80.82 | 84.25 | 80.39 | 81.57 |
| SimCSE-LLaMA2-7B ⋆ | 78.39 | 89.95 | 84.80 | 88.50 | 86.04 | 87.86 | 81.11 | 85.24 |
| AnglE-BERT | 75.09 | 85.56 | 80.66 | 86.44 | 82.47 | 85.16 | **81.23** | 82.37 |
| AnglE-LLaMA2-7B | **79.00** | **90.56** | **85.79** | **89.43** | **87.00** | **88.97** | 80.94 | **85.96** |

Table 2: Text embedding performance on STS tasks. We report the Spearman's correlation $\rho \times 100$ of the "all" setting computed by SentEval. Supervised LLaMA-based models are fine-tuned using the LoRA (Hu et al., 2021) technique. Results marked with † are obtained from (Reimers & Gurevych, 2019), while results marked with ‡ are retrieved from (Gao et al., 2021). Additionally, results marked with ⋆ denote our own implementation using official code. For the remaining baselines, we refer to the corresponding original papers to obtain their results. The T-test indicates that the improvements from AnglE to SimCSE are significant with a $p$-value of $3.71\%$.

the small-scale training set, as there are not enough samples for SimCSE to effectively learn representations. Furthermore, the datasets only provide pair-supervised data, namely $(x, x^+)$ or $(x, x^-)$, which prevents SimCSE from utilizing its hard negative objective that relies on triple-supervised data $(x, x^+, x^-)$. This limitation might affect its performance. On the other hand, AnglE consistently outperforms SBERT, achieving an absolute gain of $5.52\%$. This can support the idea that angle-optimized text embedding can mitigate the negative impact of the cosine function, resulting in better performance. Furthermore, we explore applying the long text model $RAN_{base}$ (86M parameters) (Li et al., 2023) as the backbone to test the performance on long text. The results show that AnglE-BERT outperforms AnglE-RAN across all short text datasets. This advantage might be attributed to the larger parameter size of BERT and its proficiency in handling short texts. However, we observe a remarkable shift in long-text STS. AnglE-RAN outperforms AnglE-BERT in this scenario, suggesting that AnglE-RAN can handle long texts well despite having fewer parameters.

| Model | MRPC | STS-B | QQP | QNLI | GitHub Issues. | Avg. |
|---|---|---|---|---|---|---|
| | test | test | validation | validation | test | |
| SimCSE-BERT | 48.13 | 76.27 | 65.84 | 33.00 | 60.38 | 56.72 |
| SBERT | 46.19 | 84.67 | 73.80 | 65.98 | 69.50 | 68.03 |
| AnglE-RAN | 58.70 | 80.23 | 74.87 | 63.04 | **71.25** | 69.62 |
| AnglE-BERT | **62.20** | **86.26** | **76.54** | **72.19** | 70.55 | **73.55** |

Table 3: Results on the STS tasks. All baseline results are our implementation using the official code. Spearman's correlation ($\rho \times 100$) serves as the reported metric.

In short, this evidence suggests AnglE's superiority in transfer and non-transfer settings, its ability to produce high-quality text embeddings, and its robustness and adaptability to different backbones.

## 4.4 ABLATION STUDY

To gain a deeper understanding of AnglE, we conducted an ablation study examining different objectives and their effects. The results in table 4 indicate that AnglE shows improved performance with all three objectives. In particular, we observe that AnglE experiences a greater drop in performance without the angle objective than without the in-batch negative (ibn) objective. This suggests that angle optimization is more important than ibn in improving text embedding. Additionally, we find that using the angle objective alone yields performance close to that of using the cosine objective alone, demonstrating the effectiveness of angle optimization. We also evaluated five different pooling strategies and found that the "cls" strategy performed the best. Finally, we compared the ibn with/without identical sentence pair (ISP) detection and found that ibn without ISP detection has about $0.18\%$ performance drop than with. This indicates that ibn with ISP detection is effective.

| Model | Spearman's Correlation |
|---|---|
| *Objective* | |
| AnglE-BERT-all | **86.26** |
| - w/o ibn | 86.00 |
| - w/o angle | 85.30 |
| only cosine | 85.28 |
| only ibn | 72.48 |
| only angle | 85.15 |
| *Pooling Strategy* | |
| cls | **86.26** |
| cls-last-avg | 85.81 |
| last-avg | 84.15 |
| last-max | 79.76 |
| first-last-avg | 81.99 |

Table 4: Ablation study of AnglE. The results are Spearman's correlations on the STS-B test set.

Table 5: Results of unsupervised and LLM supervised models on the STS-B test set. For ChatGPT, LLaMA, and ChatGLM, we use the gpt-turbo-3.5, 7B LLaMA2, and 6B ChatGLM, respectively.

| Model | Spearman's |
|---|---|
| *Unsupervised Models* | |
| SimCSE-BERT | 76.85 |
| ConSERT-BERT | 73.97 |
| DiffCSE-BERT | 80.59 |
| *LLM-supervised Models* | |
| AnglE-BERT + ChatGPT | 81.52 |
| AnglE-BERT + LLaMA | 79.29 |
| AnglE-BERT + ChatGLM | 81.11 |
| AnglE-BERT + Ensemble | **82.01** |

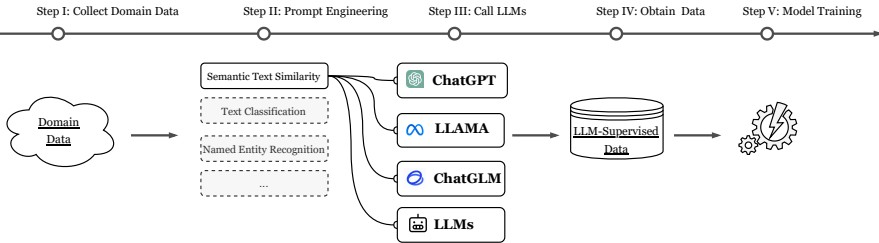

Figure 4: The procedures of the LLM-supervised learning. For the STS task, we use the prompt "*You are a highly smart same-meaning/opposite-meaning sentence-generating system, your job is to generate {size} synonymous/antonym sentences of a given input sentence. Input sentence: {text}. Output:*" to generate positive/negative pairs. {size} and {text} are placeholders for the generated size and the input text, respectively.

## 4.5 DISCUSSION AND ANALYSIS

**Discussion of LLM-supervised Learning**    AnglE, a supervised learning model, must be trained on labeled data. However, the limited availability of domain-supervised data poses a challenge in real-world applications. To overcome this problem, we propose LLM-supervised learning. This approach applies LLMs as data annotators (Schick & Schütze, 2021; Meng et al., 2022; Ye et al., 2022) to label the pseudo-supervised data for AnglE training. Figure 4 outlines the procedures involved in LLM-supervised learning. This study compares the LLM-supervised AnglE and unsupervised contrastive learning models. To simulate domain application, we extract all "sentence1" texts from the STS-B train set and employ LLM-supervised learning to train the AnglE model. Table 5 shows the results on the STS-B test set. It is evident from the results that LLM-supervised AnglE performs bet-

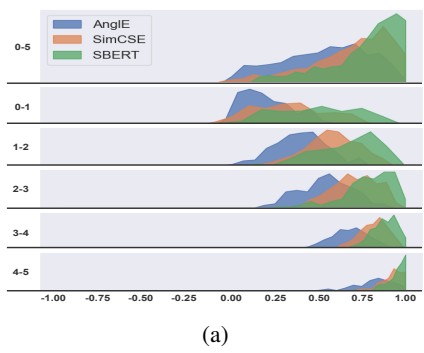 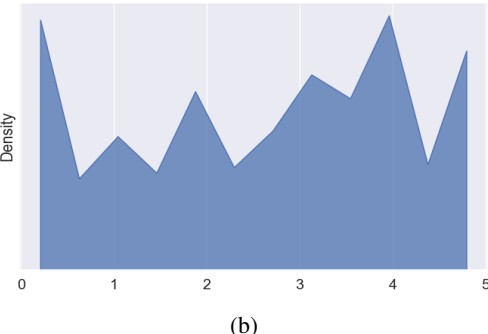

(a)                    (b)

Figure 5: (a) Density plots of cosine similarities between sentence pairs in the STS-B test set. The pairs have been categorized into 6 groups, reflecting the ground truth ratings (where higher ratings indicate a higher degree of similarity), visually represented on the y-axis. The x-axis represents the cosine similarity. (b) Density plots of golden scores between sentence pairs in the STS-B test set.

ter than unsupervised contrastive baselines, and the ensemble of LLMs shows the best results. This evidence suggests the effectiveness of LLM-supervised learning and indicates that it can alleviate the domain-supervised data scarcity problem.

**Discussion of Text Retrieval**    We also evaluate the performance of the text retrieval task by experimenting on the test split of the flickr30k dataset (Young et al., 2014). This dataset consists of five caption texts for each photo, and these texts are similar to each other. We use the first caption text vector to retrieve the top 5 similar sentences using faiss [5]. The strict accuracy [6] of AnglE, SimCSE (supervised), and SBERT are $12.9\%$, $10.4\%$, and $5.2\%$, respectively. This evidence indicates the effectiveness of using AnglE for the retrieval task.

**Analysis of Text Embedding Distribution**    Figure 5a depicts the density plots of the cosine similarities between sentence pairs in the STS-B test set to provide an intuitive visualization of the text embedding quality. Figure 5b displays the golden scores for the same sentence pairs. Analyzing the overall density of cosine similarities, we find that the AnglE distribution resembles the golden distribution more closely than SimCSE (supervised) and SBERT's. Figure 5b illustrates a peak in the 0-1 range; however, only AnglE shows a distinct peak in this range in Figure 5a. Also, Figure 5b portrays a higher peak around $4$ than around $4.8$ in the 4-5 range, only AnglE demonstrates this feature properly in Figure 5a. Notably, the 0-1 and 4-5 ranges in Figure 5b represent two saturation zones of the cosine function. This evidence suggests that AnglE can mitigate the negative effect of the saturation zone. In conclusion, we can confidently assert that AnglE produces better text embeddings with a cosine similarity density closely resembling the actual distribution than baselines.

## 5    CONCLUSION AND FUTURE WORK

In this paper, we have presented a novel text embedding model called AnglE, which optimizes the angle difference in complex space to overcome the adverse impact of the saturation zone of the cosine function, thereby improving text embeddings. To comprehensively evaluate the STS tasks, we have introduced the GitHub Issues Similarity Dataset to evaluate model performance on the long-text STS task. Furthermore, we have proposed an LLM-supervised learning method to cope with the scarcity of domain-supervised data. Extensive experimental results have demonstrated that AnglE outperforms baselines, indicating that AnglE can handle both short and long-text STS tasks and work effectively in various scenarios. In future work, we plan to explore the application of AnglE in real-world scenarios and provide further insights into AnglE.

---

[5]https://github.com/facebookresearch/faiss

[6]Strict accuracy means only all the top five sentences retrieved are equal to the five reference sentences considered correct

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

## A  APPENDIX

### A.1  RELATED WORK

Table 6 provides an overview of the similarity measurements and learning algorithms utilized in popular text embedding models. It is evident that cosine similarity is widely adopted as the measurement metric across most models. This observation underscores the importance of our proposed model, AnglE, in addressing the saturation zone issues associated with cosine similarity. By introducing AnglE, we aim to overcome the limitations posed by the saturation zone and improve the text embeddings.

| Model | Similarity Measurement | Learning Algorithm |
|---|---|---|
| BERT-flow (Li et al., 2020) | N/A | Normalizing Flow |
| BERT-whitening (Su et al., 2021) | N/A | Whitening |
| SBERT (Reimers & Gurevych, 2019) | cosine | Classification/Regression |
| SimCSE (Gao et al., 2021) | cosine | Contrastive Learning |
| ConSERT (Yan et al., 2021) | cosine | Contrastive Learning |
| DiffCSE (Chuang et al., 2022) | cosine | Contrastive Learning |

Table 6: The similarity measurements and the learning algorithms of widely-used text embedding models.

### A.2  COMPLEX SPACE DIVISION IN POLAR COORDINATES

Consider two complex numbers, $\mathbf{z} \in \mathbb{C}$ and $\mathbf{w} \in \mathbb{C}$, expressed in polar coordinates as follows:

$$
\begin{aligned}
\mathbf{z} &= \mathbf{r}_z \angle \theta_z = \mathbf{r} e^{i\theta_z} \\
\mathbf{w} &= \mathbf{r}_w \angle \theta_w = \mathbf{r} e^{i\theta_w}
\end{aligned}
\tag{9}
$$

The division of $\mathbf{z}$ and $\mathbf{w}$ is as follows:

$$
\begin{aligned}
\frac{\mathbf{z}}{\mathbf{w}} &= \left( \frac{\mathbf{r}_z}{\mathbf{r}_w} \right) e^{i\theta_z - i\theta_w} \\
&= \left( \frac{\mathbf{r}_z}{\mathbf{r}_w} \right) \angle (\theta_z - \theta_w)
\end{aligned}
\tag{10}
$$

In this manner, we have established the division of these complex numbers $\mathbf{z}$ and $\mathbf{w}$. The outcome is another complex number whose magnitude corresponds to the ratio of the magnitudes of $\mathbf{z}$ and $\mathbf{w}$, while its angle represents the difference between the angles of $\mathbf{z}$ and $\mathbf{w}$.

### A.3  DIVISION RULE IN COMPLEX SPACE

The division of two complex numbers $\mathbf{z} = \mathbf{a} + \mathbf{b}i \in \mathbb{C}$ and $\mathbf{w} = \mathbf{c} + \mathbf{d}i \in \mathbb{C}$ is as follows:

$$
\begin{aligned}
\frac{\mathbf{z}}{\mathbf{w}} &= \frac{\mathbf{a} + \mathbf{b}i}{\mathbf{c} + \mathbf{d}i} \\
&= \frac{(\mathbf{a} + \mathbf{b}i)(\mathbf{c} - \mathbf{d}i)}{(\mathbf{c} + \mathbf{d}i)(\mathbf{c} - \mathbf{d}i)} \\
&= \frac{(\mathbf{ac} + \mathbf{bd}) + (\mathbf{bc} - \mathbf{ad})i}{\mathbf{c}^2 + \mathbf{d}^2},
\end{aligned}
\tag{11}
$$

where $\mathbf{a}, \mathbf{b}, \mathbf{c}, \mathbf{d} \in \mathbb{R}$.

### A.4  LIST OF OPEN-SOURCE PROJECTS IN GITHUB ISSUE SIMILARITY DATASET

We collected GitHub issues via the official GitHub API from the following popular 55 repositories.

| tensorflow/tensorflow | pytorch/pytorch | huggingface/transformers |
| keras-team/keras | freeCodeCamp/freeCodeCamp | vuejs/vue |
| facebook/react | angular/angular | elastic/elasticsearch |
| numpy/numpy | scikit-learn/scikit-learn | pandas-dev/pandas |
| psf/requests | scipy/scipy | matplotlib/matplotlib |
| scrapy/scrapy | opencv/opencv | bumptech/glide |
| spring-projects/spring-framework | apache/dubbo | apache/superset |
| apache/airflow | apache/druid | apache/shardingsphere |
| kubernetes/kubernetes | flutter/flutter | google/jax |
| twbs/bootstrap | axios/axios | microsoft/vscode |
| sqlalchemy/sqlalchemy | mwaskom/seaborn | microsoft/TypeScript |
| microsoft/PowerToys | microsoft/terminal | microsoft/playwright |
| symfony/symfony | babel/babel | electron/electron |
| denoland/deno | mrdoob/three.js | mui/material-ui |
| webpack/webpack | atom/atom | vercel/next.js |
| ansible/ansible | npm/cli | pallets/flask |
| tiangolo/fastapi | DefinitelyTyped/DefinitelyTyped | celery/celery |
| neo4j/neo4j | rust-lang/rust | JuliaLang/julia |
| golang/go | | |

## A.5 HYPER-PARAMETERS

For a fair comparison, we follow the widely adopted baseline SimCSE Gao et al. (2021) to set the random seed $42$. We determined the values of $w_1$, $w_2$, and $w_3$ by grid searching for each dataset. For the transfer setting in the main experiment, we set the max length to $128$ and tested different batch sizes, including $8$, $16$, $32$, and $50$. Our results indicate that larger batch sizes yield better results. We used the NVIDIA 3090 Ti GPU to run our experiments. However, due to the limited GPU memory, we could not exceed a maximum batch size of $50$ in the main experiment. Therefore, we believe there is still room for improvement in the performance of AnglE. In future research, we intend to explore the effects of larger batch sizes using more advanced GPUs, such as the NVIDIA A100.

## A.6 IMPLEMENTATION OF ANGLE DIFFERENCE IN COMPLEX SPACE

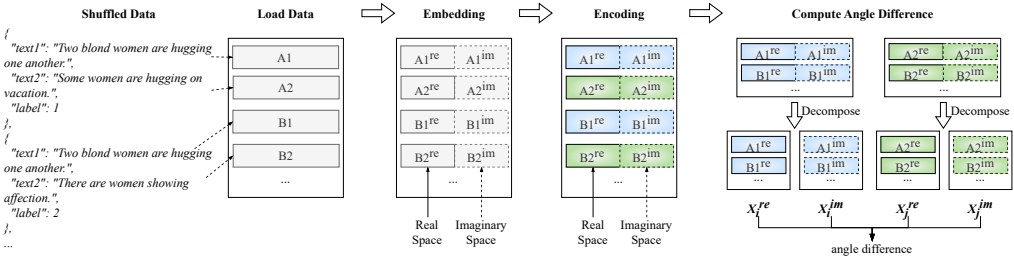

Figure 6: The procedure of computing angle difference.

Different from existing works, such as SBERT Reimers & Gurevych (2019), which use the Siamese architecture, we use the single encoder architecture. Thus, our input data is different from theirs. Our training batch data are prepared as follows:

$$\mathbf{X} = [\mathcal{A}_1; \mathcal{A}_2; \mathcal{B}_1; \mathcal{B}_2; ...; \mathcal{N}_1; \mathcal{N}_2] \in \mathbb{R}^{2B \times L \times 2D}, \tag{12}$$

where $\mathcal{A}_1$ and $\mathcal{A}_2$ are input vectors for the texts $A_1$ and $A_2$. They are supervised pairs with a positive or negative label depending on their supervised label. $2B$ is the batch size, $L$ is the padded sequence length, and $2D$ is the embedding size. We then pass it to the encoder to obtain the contextual representation as follows:

$$\mathbf{O} = \text{encoder}(\mathbf{X}) = [\mathbf{O}_{A_1}; \mathbf{O}_{A_2}; \mathbf{O}_{B_1}; \mathbf{O}_{B_2}; ...; \mathbf{O}_{N_1}; \mathbf{O}_{N_2}] \in \mathbb{R}^{2B \times L \times 2D}. \tag{13}$$

As we know, in a complex number $c = a+bi \in \mathbb{C}$, or $c = (a, b) \in \mathbb{C}$, $a$ is a real number representing the real part, $b$ is also a real number denoting the imaginary part, and $i = \sqrt{-1}$ represents the

imaginary unit. However, in the field of NLP, text is typically represented in vector space using techniques like Word Embedding (Hinton, 1984). Representing text in complex numbers is not a reasonable approach. Instead, we represent text using a complex vector, i.e., $\mathbf{c} = (\mathbf{a}, \mathbf{b}) \in \mathbb{C}$, where $\mathbf{c} = [c_1, c_2, ..., c_n]$, $\mathbf{a} = [a_1, a_2, ..., a_n]$, and $\mathbf{b} = [b_1, b_2, ..., b_n]$. Obviously, the calculation laws of complex numbers can still be applied to complex vectors. For example, the addition law in complex numbers, $(a, b) + (p, q) = (a+p, b+q)$, also applies to complex vectors, $(\mathbf{a}, \mathbf{b}) + (\mathbf{p}, \mathbf{q}) = [(a_1 + p_1, b_1 + q_1), (a_2 + p_2, b_2 + q_2), ..., (a_n + p_n, b_n + q_n)]$.

Notably, our embedding size is $2D$ instead of $D$. This is because our embedding includes two halves: the first half with size $D$ represents the real part $\mathbf{X}^{re}$, and the second half with size $D$ represents the imaginary part $\mathbf{X}^{im}$. Therefore, the embedding is presented as $[\mathbf{X}^{re}; \mathbf{X}^{im}] = [a_1, a_2, ..., a_D; b_1, b_2, ..., b_D]$. This representation strategy is sparked by (Trouillon et al., 2016; Sun et al., 2019), which decompose the entity embedding into the real part and imaginary part for the knowledge graph embeddings learning.

To compute the angle difference, we need two complex vectors. As mentioned earlier, we load the supervised pairs in batch data. After encoding, we decompose the output through tensor slicing to obtain $\mathbf{X}_i = \mathbf{O}[:: 2] = [\mathbf{O}_{A_1}; \mathbf{O}_{B_1}; ...; \mathbf{O}_{N_1}] \in \mathbb{R}^{B \times L \times 2D}$ and $\mathbf{X}_j = \mathbf{O}[1 :: 2] = [\mathbf{O}_{A_2}; \mathbf{O}_{B_2}; ...; \mathbf{O}_{N_2}] \in \mathbb{R}^{B \times L \times 2D}$, where $[::]$ is the tensor slicing operation. Next, as discussed before, we can easily obtain the real and imaginary parts of $\mathbf{X}_i$ and $\mathbf{X}_j$ by decomposing them into two halves. The real part and imaginary part are $\mathbf{a} = \mathbf{X}_i^{re} \in \mathbb{R}^{B \times L \times D}$ and $\mathbf{b} = \mathbf{X}_i^{im} \in \mathbb{R}^{B \times L \times D}$ for $\mathbf{X}_i$, and $\mathbf{c} = \mathbf{X}_j^{re} \in \mathbb{R}^{B \times L \times D}$ and $\mathbf{d} = \mathbf{X}_j^{im} \in \mathbb{R}^{B \times L \times D}$ for $\mathbf{X}_j$. We denote $\mathbf{z} = (\mathbf{a}, \mathbf{b}) \in \mathbb{C}$ and $\mathbf{w} = (\mathbf{c}, \mathbf{d}) \in \mathbb{C}$. We then compute the angle difference of pairs $(A_1, A_2)$, $(B_1, B_2)$, and others, following Equation 4, 5, and 6. Figure 6 depicts the procedure of computing the angle difference.

