# OpenReview forum: "Angle-optimized Text Embeddings"
_ICLR.cc/2024/Conference — ICLR 2024 Conference Withdrawn Submission_

### Official Review · Reviewer_7GFK · 2023-10-30

**Soundness:** 3 good
**Presentation:** 3 good
**Contribution:** 2 fair
**Rating:** 5
**Confidence:** 4

**Summary:**

This paper notices that existing text embedding models mainly use cosine function as a part of the objective function, but cosine function has a saturation zone, which may cause gradient vanishing problem and influence the quality of text embeddings. To mitigate this problem, this paper proposes to evaluate the angle difference between two text embeddings for optimization. Experiments on variable lengths of text datasets, including a newly introduced long-text dataset, are conducted to evaluate the performance of the proposed model.

**Strengths:**

1. This paper identifies an interesting research question, the gradiant vanishing problem appearing at the saturation zone of cosine function influences the quality of text embeddings.

2. The proposed solution of using angle difference for optimization is orginal and novel.

3. Experiments on semantic textual similarity task are sufficiently conducted.

**Weaknesses:**

Despite an appealing motivation and an interesting solution, I still have the following concerns:

1. From my point of view, the only technical contribution of this paper is to design how to evaluate angle difference. This contribution is indeed interesting, but is a bit superficial and insufficient for a long research paper of ICLR standard. I expect authors to propose more __insightful__ designs to better solve the gradient vanishing problem.

2. The explanation of why saturation zone in cosine function influences text embedding learning is not clearly written at the Introduction section. Authors are suggested to explain more about the meaning of saturation zone and why it causes gradient vanishing problems.

3. Usually we encourage authors to conduct the same experiment multiple times and report both mean and standard deviation, in order to verify that the proposed model indeed significantly outperforms baselines. However, I see mean but not standard deviation in the paper.

**Questions:**

1. Authors use absolute value at Eq. 6. But absolute function in pytorch or tensorflow is not differentiable, how do authors deal with error backpropagation for absolute function?

---

> ### Author Response · Authors · 2023-11-11
> **Response to Reviewer 7GFK [1/2]**
>
> We want to thank the reviewer for the professional and constructive comments.
>
> > **Weakness 1:** From my point of view, the only technical contribution of this paper is to design how to evaluate angle difference. This contribution is indeed interesting, but is a bit superficial and insufficient for a long research paper of ICLR standard. I expect authors to propose more insightful designs to better solve the gradient vanishing problem.
>
> > **Weakness 2:** The explanation of why saturation zone in cosine function influences text embedding learning is not clearly written at the Introduction section. Authors are suggested to explain more about the meaning of saturation zone and why it causes gradient vanishing problems.
>
> **Answer:** Thanks for your suggestions. We have updated the paper to include more insight and discussions in the Introduction Section. You can read the latest revision. We also provide an explanation in the following comments.
>
> It is important to note that functions with saturation zones, such as sigmoid and tanh, can lead to the problem of gradient vanishing [1]. Of course, cosine also has this issue. This means that if two points $x_1$ and $x_2$ fall within the saturation zone, the outputs $f(x_1)$ and $f(x_2)$ will be very close to each other,  where $f(*)$ denotes the neural networks. As a result, the gradient  $\Delta = f(x_1) - f(x_2) \approx 0$ will cause the gradient to vanish during backpropagation. We have illustrated this issue in Figure 1.
>
> Many STS datasets, such as MNLI and SNLI, provide three supervised labels: entailment, neutral, and contradiction. In most cases, the boundary between neutral and entailment is vague, as some neutral pairs may resemble entailment pairs.
> For instance, consider the following cases from SNLI: (1) the text pair of \textit{Two blond women hug each other.} and \textit{Some women hug each other on vacation.} are labeled as neutral, and (2) the text pair of \textit{Two blonde women hugging.} and \textit{There are women showing affection.} are labeled as entailment. The texts in the neutral pair appear very similar (a subtle difference) to each other and may fall within the saturation zone of cosine, resulting in gradient vanishing. This can confuse the model and lead to misidentification of the neutral pair as entailment.
> Additionally, many other STS datasets, such as MRPC and QQP, provide binary labels representing dissimilar ($0$) and similar ($1$), which naturally fall within the saturation zone of the cosine function.
>
> Capturing such subtle differences poses a challenge for existing text embedding models. To the best of our knowledge, no existing work has investigated this issue. To address this issue, we propose a fresh angle optimization in complex space. Optimizing the angle difference in complex space is intuitive and practical for text embedding because if the angle difference between two text embeddings is smaller, it means that the two text embeddings are closer to each other in the complex space, i.e., their similarity is larger. Other manifold-based approaches also might help, but they are complicated and have relatively higher complexity, making the model inefficient. Efficiency is essential for text embedding learning, especially in the era of LLMs. Because LLMs have large-scale parameters and are hard to finetune. Compared to them, optimizing angle differences in complex space is more practical and efficient.
>
> ---
>
> **Reference:**
>
> [1] Roodschild M, Gotay Sardiñas J, Will A. A new approach for the vanishing gradient problem on sigmoid activation[J]. Progress in Artificial Intelligence, 2020, 9(4): 351-360.

---

> > ### Author Response · Authors · 2023-11-21
> > **Kind reminder to look at the authors' reply**
> >
> > Dear Reviewer 7GFK:
> >
> > We thank you for the precious review time and valuable comments. We have provided corresponding responses with elaborate discussions on the issues you have raised. We hope to talk more with you about whether or not your concerns have been taken care of appropriately. Please let us know if you have additional questions or ideas for improvement.
> >
> > Looking forward to your reply.
> >
> > Authors.

---

> ### Author Response · Authors · 2023-11-14
> **Response to Reviewer 7GFK [2/2]**
>
> ---
>
> > **Weakness 3:** Usually we encourage authors to conduct the same experiment multiple times and report both mean and standard deviation, in order to verify that the proposed model indeed significantly outperforms baselines. However, I see mean but not standard deviation in the paper.
>
> **Answer:** Thanks for your suggestion. We think this suggestion can make our experiment more convincing. The reason we did not report the mean and standard deviation is that most STS baselines, including ours, follow SimCSE, a famous baseline in this task, to set a random seed = 42 for reproducibility. It is worth noting that after setting the random seed, the deviation difference is negligible, almost zero, in multiple runs. Notably, we have reported the p-value for the marginal results; the p-value indicates that our improvement is significant. We will add the mean and standard deviation in the camera-ready version.
>
> ---
>
> > **Question 1:** Authors use absolute value at Eq. 6. But absolute function in pytorch or tensorflow is not differentiable, how do authors deal with error backpropagation for absolute function?
>
> **Answer:** The absolute value operation $abs(*)$ is indeed a commonly used operation in deep learning. It is utilized in various loss functions, such as the Triplet Margin Loss [2], as well as in normalizations like L1 normalization. In our proposed angle loss, we also employ the absolute value operation.
>
> We think the non-differentiating is not a significant issue in deep learning frameworks like PyTorch and TensorFlow. Because even the widely-used activation function **ReLU is also non-differentiable**. The situation of $abs(*)$ is similar to ReLU. They are not differentiable at a singular point, $x=0$. Nevertheless, we can still use what is known as sub-derivatives [3] in the backpropagation algorithm.
>
> ---
>
> Finally, we would like to claim our contribution again. We have investigated the gradient vanishing problem caused by the cosine saturation zone. To the best of our knowledge, no existing works have investigated this issue before. We have proposed a fresh approach to address this issue by introducing angle optimization in complex spaces. Extensive experimental results show that we achieve SOTA performance on STS tasks, which verifies the effectiveness of our proposed model.
>
> Thank you once again for your valuable feedback!
>
> ---
>
> **Reference:**
>
> [2] Balntas V, Riba E, Ponsa D, et al. Learning local feature descriptors with triplets and shallow convolutional neural networks[C]//Bmvc. 2016, 1(2): 3.
>
> [3] https://en.m.wikipedia.org/wiki/Subderivative?wprov=sfla1

---

> > ### Author Response · Authors · 2023-11-15
> > **Response to Reviewer 7GFK [Updated Results]**
> >
> > Dear reviewer 7GFK,
> >
> > Following your advice, we have conducted multiple runs.  Here are the average Spearman’s correlation of AnglE-BERT in five runs in the transfer STS settings:
> >
> > | Run | Average Spearman’s correlation |
> > |:---:|:------------------------------:|
> > | 1   |            82.37361            |
> > | 2   |            82.37223            |
> > | 3   |            82.36981            |
> > | 4   |            82.36745            |
> > | 5   |             82.38430            |
> >
> > The mean value is calculated to be 82.37348, and the standard deviation is 0.006490023112436512.
> >
> > We will revise our paper to report this mean and standard deviation value.

---

### Official Review · Reviewer_NPkC · 2023-10-31

**Soundness:** 2 fair
**Presentation:** 3 good
**Contribution:** 2 fair
**Rating:** 5
**Confidence:** 4

**Summary:**

To overcome the negative impact of vanishing gradients caused by the cosine optimization function, this paper proposed a novel angle-optimized target to improve the quality of text embeddings. Moreover, this paper conducted extensive experiments to prove the effectiveness of the proposed method. Meanwhile, this paper also developed a novel long-text STS dataset to support the community.

**Strengths:**

1.	This paper proposed a novel angle-optimized target to enhance the learning ability of contrastive learning-based representation learning models, which tried to alleviate the problem of vanishing gradients.
2.	This paper developed a novel long-text STS dataset to better evaluate the performance of representation learning models.
3.	This paper also explored LLM-based supervised data generation and contrastive learning, which is very interesting.

**Weaknesses:**

1.	First of all, the authors argued that gradient vanishing problem is caused by the saturation zones in cosine functions in the optimization target. However, as far as I know, the gradient vanishing problem is mainly due to the deep structure. The saturation zones can be used to prove the high similarity between sentences. Therefore, the motivation of this paper is not so convincing. More explanations are needed.
2.	Second, the authors focused on contrastive learning target, which limits the application range of the proposed method. The authors should provide more evidence to demonstrate the effectiveness of their method since their main contribution is adding an additional target in contrastive loss.
3.	Third, the related work in this paper is not sufficient enough. More content should be cited, such as different contrastive loss designs, sentence similarity measurement designs, etc.

**Questions:**

N/A

---

> ### Author Response · Authors · 2023-11-11
> **Response to Reviewer NPkC [1/2]**
>
> We want to thank the reviewer for the professional and constructive comments.
>
> > **Weakness 1:**  First of all, the authors argued that gradient vanishing problem is caused by the saturation zones in cosine functions in the optimization target. However, as far as I know, the gradient vanishing problem is mainly due to the deep structure. The saturation zones can be used to prove the high similarity between sentences. Therefore, the motivation of this paper is not so convincing. More explanations are needed.
>
> **Answer:** Thanks for your suggestions. We have updated our paper to include more discussion. You can read it in our paper's latest revision or the following comments.
>
> It is important to note that functions with saturation zones, such as sigmoid and tanh, can lead to the problem of gradient vanishing [1], regardless of the depth of the network. Of course, cosine also has this issue. This means that if two points $x_1$ and $x_2$ fall within the saturation zone, the outputs $f(x_1)$ and $f(x_2)$ will be very close to each other, where $f(*)$ denotes the neural networks, it can be shallow or deep. As a result, the gradient  $\Delta = f(x_1) - f(x_2) \approx 0$ will cause the gradient to vanish during backpropagation. We have illustrated this issue in Figure 1.
>
> Many STS datasets, such as MNLI and SNLI, provide three supervised labels: entailment, neutral, and contradiction. In most cases, the boundary between neutral and entailment is vague, as some neutral pairs may resemble entailment pairs.
> For instance, consider the following cases from SNLI: (1) the text pair of \textit{Two blond women hug each other.} and \textit{Some women hug each other on vacation.} are labeled as neutral, and (2) the text pair of \textit{Two blonde women hugging.} and \textit{There are women showing affection.} are labeled as entailment. The texts in the neutral pair appear very similar (subtle differences) to each other and may fall within the saturation zone of cosine, resulting in gradient vanishing. This can confuse the model and **lead to the misidentification of the neutral pair as entailment**.
>
> Capturing such subtle differences poses a challenge for existing sentence embedding models. To the best of our knowledge, no existing work has investigated this issue. To address this issue, we propose a fresh angle optimization in complex space. Optimizing the angle difference in complex space is intuitive and practical for text embedding because if the angle difference between two text embeddings is smaller, it means that the two text embeddings are closer to each other in the complex space, i.e., their similarity is larger. Our extensive experimental results have demonstrated the effectiveness of our proposed model.
>
> ---
>
> > **Weakness 2:** Second, the authors focused on contrastive learning target, which limits the application range of the proposed method. The authors should provide more evidence to demonstrate the effectiveness of their method since their main contribution is adding an additional target in contrastive loss.
>
> **Answer:** In our ablation study, we have demonstrated the effectiveness of our proposed model. In particular, we observed that AnglE experienced a greater performance degradation without the angle objective than without the in-batch negative (ibn) objective, i.e., the contrastive objective. This suggests that angle optimization is important for improving text embedding. In addition, we find that **using the angle objective alone yields better performance than that using the ibn (contrastive) objective alone**. From the ablation study, we can also see that the angle optimization can mitigate the negative effects of the cosine function in the cosine objective (as shown in Equation (1), which is different from the contrastive learning). This evidence demonstrates the effectiveness of angle optimization for improving different models even used alone.
>
> ---
>
> **Reference:**
>
> [1] Roodschild M, Gotay Sardiñas J, Will A. A new approach for the vanishing gradient problem on sigmoid activation[J]. Progress in Artificial Intelligence, 2020, 9(4): 351-360.

---

> ### Author Response · Authors · 2023-11-14
> **Response to Reviewer NPkC [2/2]**
>
> > **Weakness 3:** Third, the related work in this paper is not sufficient enough. More content should be cited, such as different contrastive loss designs, sentence similarity measurement designs, etc.
>
> **Answer:** Thank you for your suggestions. Actually, we have cited many widely used contrastive works proposed in recent years, such as CT-BERT, SimCSE, ConSERT, DiffCSE, PromptCSE, and others, in the related work section and experiment section. Due to space limitations, we only briefly introduced them in these sections.
>
> Following your advice, **we have presented a new table (Table 6 in Appendix section A.1) to list the similarity measurements and learning algorithms employed in various widely used text embedding models in the latest revision.** Notably, we observe that the cosine similarity is predominantly used across the majority of these models. This observation underscores the significance of our work, as our proposed model, AnglE, aims to address the negative effects of the saturation zones in cosine.
>
>
>
> ---
> Finally, we would like to claim our contribution again. We have investigated the gradient vanishing problem caused by the cosine saturation zone. To the best of our knowledge, no existing works have investigated this issue before. We have proposed a fresh approach to address this issue by introducing angle optimization in complex spaces. Extensive experimental results show that we achieve SOTA performance on STS tasks, which verifies the effectiveness of our proposed model.
>
> Thank you once again for your valuable feedback!

---

> ### Author Response · Authors · 2023-11-21
> **Kind reminder to look at the authors' reply**
>
> Dear Reviewer NPkC:
>
> We thank you for the precious review time and valuable comments. We have provided corresponding responses with elaborate discussions on the issues you have raised. We hope to talk more with you about whether or not your concerns have been taken care of appropriately. Please let us know if you have additional questions or ideas for improvement.
>
> Looking forward to your reply.
>
> Authors.

---

### Official Review · Reviewer_3nS6 · 2023-11-01

**Soundness:** 2 fair
**Presentation:** 2 fair
**Contribution:** 2 fair
**Rating:** 5
**Confidence:** 3

**Summary:**

This paper proposes a novel angle-optimized text embedding model to improve the semantic textual similarity (STS) tasks, by mitigating the vanishing gradients of cos similarity. Specifically, the authors employ a contrastive learning objective and introduce optimization in a complex space to address the saturation zone in the cosine function. Extensive experiments are conducted to show the effectiveness of the proposed method on various tasks including short-text STS, long-text STS, and domain-specific STS.

**Strengths:**

1. The proposed method of calculating similarity looks novel to me.

2. The impact of the method has the potential to be significant in many fields.

**Weaknesses:**

1. According to the paper, the motivation for introducing a complex space is to deal with the vanishing gradient of cos. In this sense, it would be great if techniques like gradient clipping and gradient normalization could be compared.

2. The writing can be improved. E.g., section 3.4 is a bit confusing to me. See my questions below.

3. I am also worried about the empirical significance. In table 2, the proposed method only improves the performance marginally (<1%) compared to SimCSE-BERT. I appreciate the effort that the p-value is reported and yet the p-value is smaller than 0.05 according to the caption of table 2.

**Questions:**

1. In section 3.4, X is decomposed into real part Xre and imaginary part Xim, both of which have dimension 1. However, in the context of contrastive learning / the use of cos similarity, X is often high dimensional. How do you decompose X? If I am not mistaken, this part is missing in the paper.

---

> ### Author Response · Authors · 2023-11-11
> **Reponse to Reviewer 3nS6 [1/2]**
>
> We want to thank the reviewer for the professional and constructive comments.
>
> > **Weakness 1:** According to the paper, the motivation for introducing a complex space is to deal with the vanishing gradient of cos. In this sense, it would be great if techniques like gradient clipping and gradient normalization could be compared.
>
> **Answer:** To the best of our knowledge, techniques such as gradient clipping or gradient normalization are mainly used to prevent **exploding** gradients [1]. However, we mainly address the gradient **varnishing** problem caused by cosine.
> Actually, gradient clipping is widely used in many STS baselines, such as SimCSE and SBERT. Following these baselines, we also adopted this technique in our training phase to avoid potential gradient explosion.
>
> ---
>
> > **Weakness 2:** The writing can be improved. E.g., section 3.4 is a bit confusing to me. See my questions below.
>
> > **Question 1:** In section 3.4, X is decomposed into real part Xre and imaginary part Xim, both of which have dimension 1. However, in the context of contrastive learning / the use of cos similarity, X is often high dimensional. How do you decompose X? If I am not mistaken, this part is missing in the paper.
>
> **Answer:** We apologize for not providing a detailed implementation of the angle objective earlier due to space limitations. We have updated our paper. You can read the latest revision in Appendix Section A.5 or the following summarized comments.
>
> Firstly, our input batch data for training is as follows:
> \begin{equation}
>     \mathbf{X} = [\mathcal{A}_1;\mathcal{A}_2;\mathcal{B}_1;\mathcal{B}_2;...; \mathcal{N}_1; \mathcal{N}_2] \in \mathbb{R}^{2B\times L\times 2D},
> \end{equation}
> where $\mathcal{A}_1$ and $\mathcal{A}_2$ are input embeddings for the texts A1 and A2.
> They are supervised pairs with a positive or negative label.
> $2B$ is the batch size, $L$ is the padded sequence length, and $2D$ is the embedding size.
> We then pass it to the encoder to obtain the contextual representation as follows:
> \begin{equation}
>     \begin{split}
>         \mathbf{O} &= \mathrm{encoder}(\mathbf{X})\\\\
>         & = [\mathbf{O}_A{_1}; \mathbf{O}_A{_2}; \mathbf{O}_B{_1}; \mathbf{O}_B{_2}; ...; \mathbf{O}_N{_1}; \mathbf{O}_N{_2}]  \in \mathbb{R}^{2B\times L\times 2D} .
>     \end{split}
> \end{equation}
>
> We extend the complex number to the complex vector (see appendix section A.5). For a complex vector $\mathbf{c} = \mathbf{a} + \mathbf{b}i$, $\mathbf{a}$ is a real vector representing the real part, $\mathbf{b}$ is also a real vector denoting the imaginary part, and $i$ represents the imaginary unit.
>
> Notably, our embedding size is $2D$ instead of $D$. This is because our embedding includes two halves: the first half with size $D$ represents the real part $\mathbf{X}^{re}$, and the second half with size $D$ represents the imaginary part $\mathbf{X}^{im}$. Therefore, the embedding is presented as $[\mathbf{X}^{re}; \mathbf{X}^{im}]$. This representation strategy follows RotatE[2], which decomposes the entity embedding into the real part and imaginary part for knowledge graph embedding learning.
>
> To compute the angle difference, we need two complex vectors. Specifically, we decompose the output through tensor slicing to obtain pairs: $\mathbf{X}_i = \mathbf{O}[::2] = [\mathbf{O}_A{_1}; \mathbf{O}_B{_1};...; \mathbf{O}_N{_1}] \in \mathbb{R}^{B\times L \times 2D}$ and $\mathbf{X}_j = \mathbf{O}[1::2] = [\mathbf{O}_A{_2}; \mathbf{O}_B{_2};...; \mathbf{O}_N{_2}]  \in \mathbb{R}^{B\times L \times 2D}$, where $[::]$ is the tensor slicing operation.
>
> Next, as discussed before, we can easily obtain the real and imaginary parts of pairs $\mathbf{X}_i$ and $\mathbf{X}_j$.
> The real part and imaginary part are $\mathbf{a} \in \mathbb{R}^{B\times L \times D}$ and $\mathbf{b}  \in \mathbb{R}^{B\times L \times D}$ for $\mathbf{X}_i$, and $\mathbf{c}  \in \mathbb{R}^{B\times L \times D} $ and $\mathbf{d}  \in \mathbb{R}^{B\times L \times D}$ for $\mathbf{X}_j$. We denote $\mathbf{z} = (\mathbf{a}, \mathbf{b}) \in \mathbb{C}$ and $\mathbf{w} = (\mathbf{c}, \mathbf{d}) \in \mathbb{C}$. Next, we can compute the angle difference of pairs ($A_1$, $A_2$),  ($B_1$, $B_2$), and others, following Equations (4), (5), and (6) in the paper.
>
> We also provide a figure (Figure 6) to illustrate intuitively the process of computing the angle difference. For more implementation details, you can read the appendix section A.5 in our latest revision paper.
>
>
> ---
>
> **Reference**
>
> [1] Zhang J, He T, Sra S, et al. Why gradient clipping accelerates training: A theoretical justification for adaptivity[J]. arXiv preprint arXiv:1905.11881, 2019.
>
> [2] rotate: knowledge graph embedding by relational rotation in complex space (ICLR19)

---

> ### Author Response · Authors · 2023-11-14
> **Reponse to Reviewer 3nS6 [2/2]**
>
> We want to thank the reviewer for the professional and constructive comments.
>
> > **Weakness 3:** I am also worried about the empirical significance. In table 2, the proposed method only improves the performance marginally (<1%) compared to SimCSE-BERT. I appreciate the effort that the p-value is reported and yet the p-value is smaller than 0.05 according to the caption of table 2.
>
> **Answer:** Although our results are marginally better than SimCSE-BERT in the transfer STS setting in terms of average scores, it is worth noting that AnglE-BERT outperforms SimCSE-BERT in 6 out of 7 STS datasets. Specifically, AnglE-BERT shows obvious improvement in STS13, STS15, STS16, STSB, and SickR datasets. The reported p-value for the average score confirms the significance of the marginal improvement in the average score.
> In addition, our updated LLaMA results show that AnglE-LLaMA outperforms SimCSE-LLaMA, achieving 6 out of 7 best scores.
>
> On the other hand, AnglE-BERT shows an **absolute improvement** over SimCSE-BERT in the non-transfer STS settings.
>
> This evidence suggests that AnglE is effective and can be applied to both transfer and non-transfer settings.
>
> ---
>
> Finally, we would like to claim our contribution again. We have investigated the gradient vanishing problem caused by the cosine saturation zone. To the best of our knowledge, no existing works have investigated this issue before. We have proposed a fresh approach to address this issue by introducing angle optimization in complex spaces. Extensive experimental results show that we achieve SOTA performance on STS tasks, which verifies the effectiveness of our proposed model.
>
>
> Thank you once again for your valuable feedback!

---

> ### Author Response · Authors · 2023-11-21
> **Kind reminder to look at the authors' reply**
>
> Dear Reviewer 3nS6:
>
> We thank you for the precious review time and valuable comments. We have provided corresponding responses with elaborate discussions on the issues you have raised. We hope to talk more with you about whether or not your concerns have been taken care of appropriately. Please let us know if you have additional questions or ideas for improvement.
>
> Looking forward to your reply.
>
> Authors.

---

> ### Comment · Reviewer_3nS6 · 2023-11-22
>
> Thank you for your response. After reading all the review comments I tend to keep my rating unchanged.

---

### Official Review · Reviewer_DPjM · 2023-11-02

**Soundness:** 3 good
**Presentation:** 2 fair
**Contribution:** 3 good
**Rating:** 6
**Confidence:** 3

**Summary:**

The paper proposes a new method called AnglE to address the vanishing gradient problem of optimizing cosine similarity in text embedding learning models. AnglE uses an angle-based optimization method to learn text embeddings in a complex space. The method is demonstrated to outperform state-of-the-art models on various semantic textual similarity (STS) tasks, including short-text STS, long-text STS, and domain-specific STS. Additionally, AnglE can be used with limited labeled data and LLM-annotated data, and it achieves competitive performance in these settings.

**Strengths:**

* The paper addresses an important issue in optimizing the cosine similarity of learning text embeddings, and the proposed method is interesting and novel.
* It introduces the GitHub Issues Similarity Dataset as a testbed for evaluating model performance on long-text STS tasks.
* The proposed method achieves promising results on a wide range of STS tasks.

**Weaknesses:**

* Some technical details are not clearly explained. For example, while the angle objective optimizes the text representations in a complex space, it's unclear how these complex vectors are obtained as the representations from language models are real vectors.
* The paper seems to have missed discussions with a few important related studies. For example, [1] addresses the gradient vanishing issue by incorporating cosine distance in learning text embeddings, [2] designs angular softmax objectives to learn visual representations. The LLM-supervised learning procedure largely follows the prompt-based training data generation paradigm in [3,4,5]. While this part is not the major contribution of the paper, it's better to reference these related works as well.

References:
- [1] “Spherical Text Embedding.” NeurIPS (2019).
- [2] “SphereFace: Deep Hypersphere Embedding for Face Recognition.” CVPR (2017).
- [3] “Generating Datasets with Pretrained Language Models.” EMNLP (2021).
- [4] “Generating Training Data with Language Models: Towards Zero-Shot Language Understanding.” NeurIPS (2022).
- [5] “ZeroGen: Efficient Zero-shot Learning via Dataset Generation.” EMNLP (2022).

**Questions:**

* Could you explain how the complex vectors are obtained exactly from the language models?

---

> ### Author Response · Authors · 2023-11-11
> **Response to Reviewer DPjM [1/2]**
>
> We want to thank the reviewer for the professional and constructive comments.
>
>
> > **Weakness1**: Some technical details are not clearly explained. For example, while the angle objective optimizes the text representations in a complex space, it's unclear how these complex vectors are obtained as the representations from language models are real vectors.
>
> > **Question1**: Could you explain how the complex vectors are obtained exactly from the language models?
>
> **Answer:** We apologize for not providing a detailed implementation of the angle objective earlier due to space limitations. We have updated our paper. You can read the latest revision in Appendix Section A.5 or the following summarized comments.
>
> Firstly, our input batch data for training is as follows:
> \begin{equation}
>     \mathbf{X} = [\mathcal{A}_1;\mathcal{A}_2;\mathcal{B}_1;\mathcal{B}_2;...; \mathcal{N}_1; \mathcal{N}_2] \in \mathbb{R}^{2B\times L\times 2D},
> \end{equation}
> where $\mathcal{A}_1$ and $\mathcal{A}_2$ are input embeddings for the texts A1 and A2.
> They are supervised pairs with a positive or negative label.
> $2B$ is the batch size, $L$ is the padded sequence length, and $2D$ is the embedding size.
> We then pass it to the encoder to obtain the contextual representation as follows:
> \begin{equation}
>     \begin{split}
>         \mathbf{O} &= \mathrm{encoder}(\mathbf{X})\\\\
>         & = [\mathbf{O}_A{_1}; \mathbf{O}_A{_2}; \mathbf{O}_B{_1}; \mathbf{O}_B{_2}; ...; \mathbf{O}_N{_1}; \mathbf{O}_N{_2}]  \in \mathbb{R}^{2B\times L\times 2D} .
>     \end{split}
> \end{equation}
>
> We extend the complex number to the complex vector (see appendix section A.5). For a complex vector $\mathbf{c} = \mathbf{a} + \mathbf{b}i$, $\mathbf{a}$ is a real vector representing the real part, $\mathbf{b}$ is also a real vector denoting the imaginary part, and $i$ represents the imaginary unit.
>
> Notably, our embedding size is $2D$ instead of $D$. This is because our embedding includes two halves: the first half with size $D$ represents the real part $\mathbf{X}^{re}$, and the second half with size $D$ represents the imaginary part $\mathbf{X}^{im}$. Therefore, the embedding is presented as $[\mathbf{X}^{re}; \mathbf{X}^{im}]$. This representation strategy follows RotatE[1], which decomposes the entity embedding into the real part and imaginary part for knowledge graph embedding learning.
>
> To compute the angle difference, we need two complex vectors. Specifically, we decompose the output through tensor slicing to obtain pairs: $\mathbf{X}_i = \mathbf{O}[::2] = [\mathbf{O}_A{_1}; \mathbf{O}_B{_1};...; \mathbf{O}_N{_1}] \in \mathbb{R}^{B\times L \times 2D}$ and $\mathbf{X}_j = \mathbf{O}[1::2] = [\mathbf{O}_A{_2}; \mathbf{O}_B{_2};...; \mathbf{O}_N{_2}]  \in \mathbb{R}^{B\times L \times 2D}$, where $[::]$ is the tensor slicing operation.
>
> Next, as discussed before, we can easily obtain the real and imaginary parts of pairs $\mathbf{X}_i$ and $\mathbf{X}_j$.
> The real part and imaginary part are $\mathbf{a} \in \mathbb{R}^{B\times L \times D}$ and $\mathbf{b}  \in \mathbb{R}^{B\times L \times D}$ for $\mathbf{X}_i$, and $\mathbf{c}  \in \mathbb{R}^{B\times L \times D} $ and $\mathbf{d}  \in \mathbb{R}^{B\times L \times D}$ for $\mathbf{X}_j$. We denote $\mathbf{z} = (\mathbf{a}, \mathbf{b}) \in \mathbb{C}$ and $\mathbf{w} = (\mathbf{c}, \mathbf{d}) \in \mathbb{C}$. Next, we can compute the angle difference of pairs ($A_1$, $A_2$),  ($B_1$, $B_2$), and others, following Equations (4), (5), and (6) in the paper.
>
> We also provide a figure (Figure 6) to illustrate intuitively the process of computing the angle difference. For more implementation details, you can read the appendix section A.5 in our latest revision paper.
>
>
> ---
> **Reference**
>
> [1] rotate: knowledge graph embedding by relational rotation in complex space (ICLR19)

---

> ### Author Response · Authors · 2023-11-14
> **Response to Reviewer DPjM [2/2]**
>
> We want to thank the reviewer for the professional and constructive comments.
>
>
> > **Weakness 2**: The paper seems to have missed discussions with a few important related studies. For example, [1] addresses the gradient vanishing issue by incorporating cosine distance in learning text embeddings, [2] designs angular softmax objectives to learn visual representations. The LLM-supervised learning procedure largely follows the prompt-based training data generation paradigm in [3,4,5]. While this part is not the major contribution of the paper, it's better to reference these related works as well.
>
> **Answer:** Thanks for pointing out these related works. We have addressed this issue in our latest revision of the paper by including the citations and discussing the differences between our approach and the referenced works in the introduction and experiment sections.
> Regarding work [1], it utilizes the marginal loss as the objective function and optimizes it through Riemannian optimization in spherical manifolds. As for work [2], they manipulate decision boundaries to produce an angular margin.
> Our work is different from theirs. We compute the angle difference in complex space and aim to address the negative effect caused by the saturation zone of cosine.
>
> For other related works on LLM-based training data generation, we have cited them in Section 4.5 in the latest revision of our paper.
>
> ---
>
> Finally, we would like to claim our contribution again. We have investigated the gradient vanishing problem caused by the cosine saturation zone. To the best of our knowledge, no existing works have investigated this issue before. We have proposed a fresh approach to address this issue by introducing angle optimization in complex spaces. Extensive experimental results show that we achieve SOTA performance on STS tasks, which verifies the effectiveness of our proposed model.
>
> Thank you once again for your valuable feedback!

---

> ### Author Response · Authors · 2023-11-21
> **Kind reminder to look at the authors' reply**
>
> Dear Reviewer DPjM:
>
> We thank you for the precious review time and valuable comments. We have provided corresponding responses with elaborate discussions on angle optimization, which we hope to address your concerns. We hope to talk more with you about whether or not your concerns have been taken care of appropriately. Please let us know if you have additional questions or ideas for improvement.
>
> Looking forward to your reply.
>
> Authors.

---

> > ### Comment · Reviewer_DPjM · 2023-11-23
> > **Response to Authors**
> >
> > I thank the authors for their response. I'm keeping my score since it was given under the expectation that the authors could provide clarifications for my questions.

---

### Author Response · Authors · 2023-11-13
**Response to all reviewers**

Dear all reviewers,

We would like to thank you for your professional and constructive comments.

We have carefully considered your feedback and made the necessary updates to our paper. You can review the latest revision to see the changes we have made. **We look forward to receiving your updated feedback.**

In this shared comment box, we would like to provide some explanations of our updates in the paper for your convenience.

1) Introduction section: We have expanded our discussion of the gradient vanishing problem associated with the cosine saturation zone. In addition, we have provided a further explanation of how our proposed model, AnglE, effectively addresses this problem.

2) Main results section: We have updated our LLaMA results. It achieved state-of-the-art performance on STS tasks at the 7B LLMs scale.

3) **New Appendix Section A.5: Due to page limitations, we were unable to provide a detailed explanation of the implementation in the previous version. We have added a new section providing a comprehensive overview of the angle difference implementation. We also provide a figure to illustrate intuitively the process of computing the angle difference.**

4) **New Appendix Section A.1 (Related Work)**: We present an overview of the similarity measurements and learning algorithms employed in various widely-used text embedding models. Notably, we observe that the cosine similarity is predominantly used across the majority of these models. This observation underscores the significance of our work, as our proposed model, AnglE, aims to address the negative effects of the saturation zones in cosine.

---

### Author Response · Authors · 2023-11-20
**We look forward to your discussion**

Dear Reviewers and ACs,

We understand the workload and time constraints you might face.
We have dedicated much effort to address your questions and concerns. **But we haven't received any feedback yet.** The discussion **DDL is Nov 22**. It is around the corner.
**We look forward to your further feedback about our responses and expect more concrete and constructive comments from you to improve our work.**


Sincerely,

Authors